

# Invariant approach to the driven Jaynes-Cummings model

Ivan Alejandro Bocanegra-Garay[1,2][*], L. Hernández-Sánchez[1], Irán Ramos-Prieto[1],
Francisco Soto-Eguibar[1] and Héctor Manuel Moya-Cessa[1]

**1** Instituto Nacional de Astrofísica Óptica y Electrónica, Calle Luis Enrique Erro No. 1,
Santa María Tonantzintla, Puebla, 72840, Mexico
**2** Departamento de Física Teórica, Atómica y Óptica,
Universidad de Valladolid, 47011 Valladolid, Spain

* ivanbocanegrag@gmail.com

## Abstract

We investigate the dynamics of the driven Jaynes-Cummings model, where a two-level atom interacts with a quantized field and both, atom and field, are driven by an external classical field. Via an invariant approach, we are able to transform the corresponding Hamiltonian into the one of the standard Jaynes-Cummings model. Subsequently, the exact analytical solution of the Schrödinger equation for the driven system is obtained and employed to analyze some of its dynamical variables.



# 1 Introduction

The Jaynes-Cummings model (JCM) is probably the most fundamental theoretical model in quantum optics [1]. It is also the simplest exactly solvable model describing the interaction between matter and electromagnetic radiation. The JCM consists of a single two-level atom interacting with a single quantized mode of the electromagnetic field in a lossless cavity, under the dipole and rotating wave approximations [2].

In the strong-coupling (or ultra-strong coupling) regime of matter-radiation interaction, when the rotating wave approximation is not valid anymore, the corresponding model is known as the Rabi model. It has been proved that the presence of the counter-rotating terms in the corresponding (Rabi) Hamiltonian is responsible for richer dynamics [3, 4]. Although numerical results have been known for a while, the Rabi model has only been recently solved by Braak, through highly sophisticated analytical techniques [5].

In turn, over the years, the JCM has been exhaustively studied, extended and generalized. Such generalizations intend to address more involved and realistic aspects of the interaction between atoms and fields, beyond the simplifications of the original model [6–9]. These include the generalized JCM (which incorporates multiple atomic levels [10–12] or field modes [13, 14]), the dispersive JCM [15], models including nonlinear effects [16–18] and losses [19, 20], among others [21–25].

The standard JCM predicts that the atom and the quantized field become entangled, ceasing to be individual systems and turning into a kind of "molecule" [26]. In fact, Alsing et al. [26] demonstrated that in order to analyze this molecule, it is necessary to probe it in some manner, and they showed that an external classical field is the natural way to do it. This leads to a new generalization of the conventional JCM, referred to as the "driven Jaynes-Cummings model" [26, 27]. In their article, Alsing et al. analyze two types of driving mechanisms; the first involves the external classical field driving the cavity mode (which was experimentally reported by Thompson [28] et al.), and the second involves the classical field driving the two-level atom. In both cases, the eigenenergies and eigenstates of the system are determined. However, we emphasize that nothing about the dynamical variables of the system (atomic inversion, average photon number, etc.) is said in [26].

Additionally, Dutra et al. [27] also analyze the scenario in which the classical field drives the atom only, discussing the necessary criteria for the model to have physical significance. They also show how the driven JCM can be transformed into the standard one, enabling the calculation of certain dynamic variables of the system.

In this study, we are interested in investigating the most general case of the driven Jaynes-Cummings model, which allows for the simultaneous excitation of both, the atom and the quantized field, by the presence of an external classical field. Our aim is to establish a methodology that enables the direct calculation of the dynamic variables of the driven system in a straightforward manner, and not presented in [26]. Therefore, obtaining the solution of the Schrödinger equation in the general driven case constitutes our main motivation and contribution. In that respect, the present work represents also a generalization of [27]. Note also that although we are studying the interaction between two fields (classical and quantum) with a two-level atom, the Hamiltonian we solve may also appear in ion-laser interactions [29–31].

The paper is organized as follows: in Section 2, a detailed description of the model under study is given, and the invariant approach is described. In Section 3, the time-dependent Hamiltonian of the driven system is related to that of the conventional JCM, by means of unitary transformations, and according to the invariant technique. In Section 4, the general solution to the Schrödinger equation for the driven Hamiltonian is obtained. By choosing specific initial conditions, for both the atom and the quantized field, the atomic inversion, the average photon number, the Mandel Q parameter, and the entropy of the system are easily

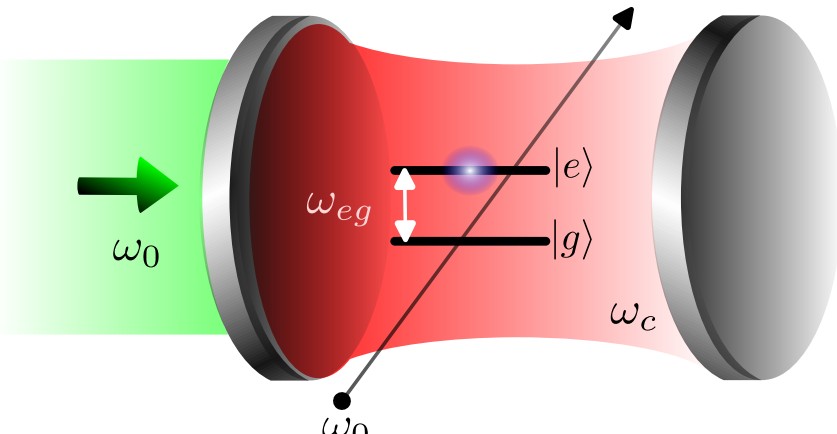

Figure 1: Scheme of a lossless cavity formed by perfectly reflecting mirrors (in shades of gray). Within the space between the mirrors, a two-level atom with a transition frequency $\omega_{eg}$ interacts with a quantized field of frequency $\omega_c$ (in shades of red). Additionally, both the atom and the quantized field, are influenced by an external classical field with frequency $\omega_0$. The classical field that drives the quantized field is represented by the thick horizontal arrow in green, while the one that impinges on the atom is depicted by a thin diagonal arrow in black.

determined (as examples of the dynamical variables) and studied from the general solution. In Section 5, the dispersive regime is considered. Finally, in Section 6 the main conclusions are presented.

## 2  The driven system and the invariant approach

Let us consider a system consisting of a two-level atom, with states denoted as $|g\rangle$ (ground state) and $|e\rangle$ (excited state), having a transition frequency $\omega_{eg}$. The atom is placed within a cavity (the reader may think of a cavity formed by perfectly reflecting mirrors) sustaining a single quantized electromagnetic field mode with frequency $\omega_c$. Additionally, an external classical field with frequency $\omega_0$ driving both the atom and the quantized field is considered. This setup is depicted in Fig. 1. Furthermore, we assume that the coupling between the cavity mode and the atom is significantly larger than the cavity damping and atomic decay rates, enabling us to neglect their effects [27]. Based on these assumptions, and in the dipole and rotating wave approximations, the time-dependent Hamiltonian describing the system can be written as [26]

$$\hat{\mathcal{H}} = \frac{\omega_{eg}}{2}\hat{\sigma}_z + \omega_c \hat{a}^\dagger \hat{a} + g\left(\hat{\sigma}_+ \hat{a} + \hat{\sigma}_- \hat{a}^\dagger\right) + \zeta\left(\hat{\sigma}_- e^{\mathrm{i}\omega_0 t} + \hat{\sigma}_+ e^{-\mathrm{i}\omega_0 t}\right) + \xi\left(\hat{a} e^{\mathrm{i}\omega_0 t} + \hat{a}^\dagger e^{-\mathrm{i}\omega_0 t}\right), \quad (1)$$

where the real parameters $g$, $\zeta$ and $\xi$ are the coupling constants between the atom and the quantized field, the external classical field and the atom, and the quantized and classical fields, respectively. As usual, the creation and annihilation operators $\hat{a}^\dagger$ and $\hat{a}$, satisfying the commutation relation $\left[\hat{a}, \hat{a}^\dagger\right] = 1$, are considered for the quantized field, while the pseudo-spin operators $\hat{\sigma}_+ = |e\rangle\langle g|$, $\hat{\sigma}_- = |g\rangle\langle e|$, and $\hat{\sigma}_z = |e\rangle\langle e| - |g\rangle\langle g|$, with the commutation relations $[\hat{\sigma}_+, \hat{\sigma}_-] = \hat{\sigma}_z$ and $[\hat{\sigma}_z, \hat{\sigma}_\pm] = \pm 2\hat{\sigma}_\pm$, describe the atomic part of the system.

It is possible to write an invariant $\hat{I}$ satisfying [32, 33]

$$\frac{d\hat{I}}{dt} = \frac{\partial \hat{I}}{\partial t} - \mathrm{i}[\hat{I}, \hat{\mathcal{H}}] = 0, \quad (2)$$

for the time-dependent Hamiltonian (1), in the form

$$\hat{I} = \frac{\hat{\sigma}_z}{2} + \hat{a}^\dagger \hat{a} + \alpha \left( \hat{a} e^{i\omega_0 t} + \hat{a}^\dagger e^{-i\omega_0 t} \right), \tag{3}$$

where $\alpha$ is a real constant to be determined. For $\hat{I}$ to fulfill (2), it is necessary that $\alpha = \zeta/g$ and $\xi = \alpha(\omega_c - \omega_0)$, constrictions that prevent the classical and quantized fields to be on resonance.

The time dependence of the invariant (3) can be eliminated by changing to a frame rotating at frequency $\omega_0$, i.e.,

$$\hat{I}_T := \hat{T} \hat{I} \hat{T}^\dagger = \frac{\hat{\sigma}_z}{2} + \hat{a}^\dagger \hat{a} + \alpha \left( \hat{a} + \hat{a}^\dagger \right), \tag{4}$$

where $\hat{T} = \exp[i\omega_0 t(\hat{n} + \hat{\sigma}_z/2)]$, with $\hat{n} = \hat{a}^\dagger \hat{a}$ the usual number operator. Furthermore, by transforming (4) with the Glauber displacement operator $\hat{D}(\alpha) = \exp\left[\alpha(\hat{a}^\dagger - \hat{a})\right]$ [34], the well-known constant of motion of the standard JCM is obtained [35]

$$\hat{I}_D := \hat{D}(\alpha) \hat{I}_T \hat{D}^\dagger(\alpha) = \frac{\hat{\sigma}_z}{2} + \hat{a}^\dagger \hat{a}. \tag{5}$$

The above result suggests that properly transforming the Hamiltonian (1), we can arrive at the (solvable) Jaynes-Cummings Hamiltonian, as we show in the next section. It is important to emphasize that if a classical field drives either only the atom or the quantum field, it is not possible to write an invariant. Furthermore, it is noteworthy also to stress that when the classical field solely drives the quantum field, even though a solution exists [26], it is actually hardly useful to study the system dynamics.

## 3 Connection between the driven and the standard JCM

The dynamics of the system associated to the Hamiltonian (1) is governed by the Schrödinger equation

$$i\frac{\partial |\psi(t)\rangle}{\partial t} = \hat{\mathcal{H}} |\psi(t)\rangle. \tag{6}$$

As we saw previously, we can move to a frame that rotates at frequency $\omega_0$. We propose $|\psi(t)\rangle = \hat{T}^\dagger |\phi(t)\rangle$; therefore, the resulting Schrödinger equation is

$$i\frac{\partial |\phi(t)\rangle}{\partial t} = \hat{\mathcal{H}}_T |\phi(t)\rangle, \tag{7}$$

with

$$\hat{\mathcal{H}}_T = \hat{T} \hat{\mathcal{H}} \hat{T}^\dagger - i\hat{T} \partial_t \hat{T}^\dagger = \Delta_c \hat{n} + \frac{\Delta_{eg}}{2} \hat{\sigma}_z + g\left(\hat{\sigma}_+ \hat{a} + \hat{\sigma}_- \hat{a}^\dagger\right) + \zeta\left(\hat{\sigma}_- + \hat{\sigma}_+\right) + \xi\left(\hat{a} + \hat{a}^\dagger\right), \tag{8}$$

where $\Delta_c = \omega_c - \omega_0$ and $\Delta_{eg} = \omega_{eg} - \omega_0$ represent the detunnings between the quantized and the classical fields, and the atomic and the classical field frequencies, respectively.

If now we perform a unitary transformation such that $|\phi(t)\rangle = \hat{D}^\dagger(\alpha)|\chi(t)\rangle$, we arrive to the Schrödinger equation

$$i\frac{\partial |\chi(t)\rangle}{\partial t} = \hat{\mathcal{H}}_D |\chi(t)\rangle, \tag{9}$$

where $\hat{\mathcal{H}}_D$ is given by

$$\begin{aligned}
\hat{\mathcal{H}}_D = \hat{D}(\alpha) \hat{\mathcal{H}}_T \hat{D}^\dagger(\alpha) = \Delta_c \hat{n} + \frac{\Delta_{eg}}{2} \hat{\sigma}_z + g\left(\hat{\sigma}_+ \hat{a} + \hat{\sigma}_- \hat{a}^\dagger\right) + \alpha(\alpha\Delta_c - 2\xi) \\
+ (\zeta - g\alpha)(\hat{\sigma}_- + \hat{\sigma}_+) + (\xi - \alpha\Delta_c)\left(\hat{a} + \hat{a}^\dagger\right).
\end{aligned} \tag{10}$$

As before, setting

$$\alpha = \frac{\zeta}{g}, \tag{11}$$

and $\Delta_c = g\xi/\zeta$, the last two terms in (10) are eliminated, and we obtain

$$\hat{\mathcal{H}}_D = \hat{D}(\zeta/g)\hat{\mathcal{H}}_T\hat{D}^\dagger(\zeta/g) = \Delta_c\hat{n} + \frac{\Delta_{eg}}{2}\hat{\sigma}_z + g\left(\hat{\sigma}_+\hat{a} + \hat{\sigma}_-\hat{a}^\dagger\right) - \zeta\xi/g. \tag{12}$$

The last term above can be ignored, as it does not play any role in the dynamics of the system, resulting in the standard Jaynes-Cummings Hamiltonian

$$\hat{\mathcal{H}}_{\text{JCM}} = \Delta_c\hat{n} + \frac{\Delta_{eg}}{2}\hat{\sigma}_z + g\left(\hat{\sigma}_+\hat{a} + \hat{\sigma}_-\hat{a}^\dagger\right). \tag{13}$$

Finally, we can move to a frame rotating at frequency $\Delta_c$, via the transformation $\hat{S} = \exp[\mathrm{i}\Delta_c t(\hat{n} + \hat{\sigma}_z/2)]$, such that $|\chi(t)\rangle = \hat{S}^\dagger|\eta(t)\rangle$. The equation to solve is then

$$\mathrm{i}\frac{\partial|\eta(t)\rangle}{\partial t} = \hat{\mathcal{H}}_{\text{I}}|\eta(t)\rangle, \tag{14}$$

with the interaction Hamiltonian

$$\hat{\mathcal{H}}_{\text{I}} = \frac{\Delta}{2}\hat{\sigma}_z + g\left(\hat{\sigma}_+\hat{a} + \hat{\sigma}_-\hat{a}^\dagger\right), \tag{15}$$

where $\Delta = \Delta_{eg} - \Delta_c = \omega_{eg} - \omega_c$.

## 4  Dynamics

The evolution operator associated to (15) is widely known [7, 8, 36, 37], and can be expressed as

$$\hat{U}_{\text{I}} = e^{-\mathrm{i}t\hat{\mathcal{H}}_{\text{I}}} = \begin{pmatrix} \hat{U}_{11}(t) & \hat{U}_{12}(t) \\ \hat{U}_{21}(t) & \hat{U}_{22}(t) \end{pmatrix}, \tag{16}$$

where

$$\hat{U}_{11}(t) = \cos\left(\hat{\Omega}_{n+1}t\right) - \mathrm{i}\frac{\Delta}{2}\frac{\sin\left(\hat{\Omega}_{n+1}t\right)}{\hat{\Omega}_{n+1}}, \tag{17a}$$

$$\hat{U}_{12}(t) = -\mathrm{i}g\hat{a}\frac{\sin\left(\hat{\Omega}_n t\right)}{\hat{\Omega}_n}, \tag{17b}$$

$$\hat{U}_{21}(t) = -\mathrm{i}g\hat{a}^\dagger\frac{\sin\left(\hat{\Omega}_{n+1}t\right)}{\hat{\Omega}_{n+1}}, \tag{17c}$$

$$\hat{U}_{22}(t) = \cos\left(\hat{\Omega}_n t\right) + \mathrm{i}\frac{\Delta}{2}\frac{\sin\left(\hat{\Omega}_n t\right)}{\hat{\Omega}_n}, \tag{17d}$$

and

$$\hat{\Omega}_n = \left(\frac{\Delta^2}{4} + g^2\hat{n}\right)^{1/2}. \tag{18}$$

Then, the solution of the initial Schrödinger equation (6) is given by

$$|\psi(t)\rangle = \hat{T}^\dagger\hat{D}^\dagger(\zeta/g)\hat{S}^\dagger\hat{U}_{\text{I}}(t)\hat{D}(\zeta/g)|\psi(0)\rangle, \tag{19}$$

since $\hat{T}(0) = \hat{S}(0) = \hat{1}$, with $\hat{1}$ the identity operator. Recall that we have set $\Delta_c = g\xi/\zeta$, thus there are only five free parameters out of the initial six parameters in the Hamiltonian (1).

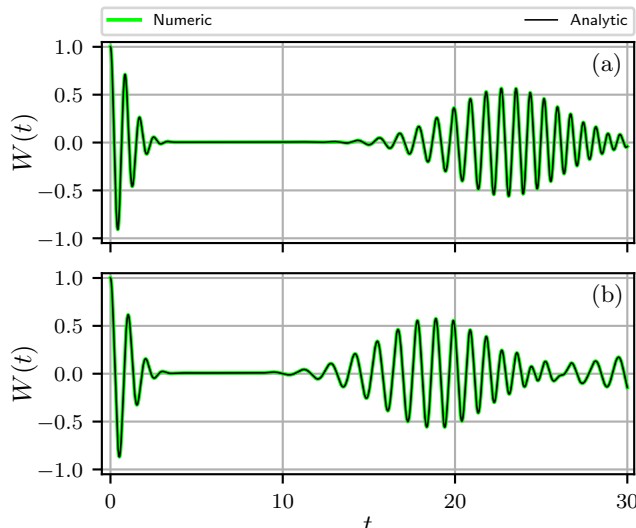

Figure 2: Atomic inversion $W(t)$ corresponding to the initial condition $|\psi(0)\rangle = |\beta, e\rangle$, using the following parameter values: $\omega_c = 0.4$, $\omega_{eg} = 0.9$, $g = 1.0$, $\zeta = 0.7$, $\xi = 0.2$, $\omega_0 = \omega_c - g\xi/\zeta$, and $\beta = \sqrt{8}$. In (a) the atomic inversion corresponding to the driven JCM is shown. In (b) the atomic inversion corresponding to the conventional JCM is displayed. The black lines represent the analytical result, while the green ones stand for the numerical solution obtained using QuTiP [38].

From now on, we set $\omega_0 = \omega_c - g\xi/\zeta$.

The general solution (19) allows to calculate and analyze the dynamical variables of the driven system, enabling also a direct comparison with the standard JCM, as shown next. For the sake of simplicity, we consider that the field is initially in a coherent state $|\beta\rangle$, where $\beta$ is an arbitrary complex number, while the atom is in the excited state $|e\rangle$; that is, our initial state will be $|\psi(0)\rangle = |\beta\rangle \otimes |e\rangle = |\beta, e\rangle$.

## 4.1 Atomic inversion

The atomic inversion $W(t)$ is a meaningful observable that indicates changes in the population distribution of atoms and contains important statistical information of the field. It is defined as the difference between the probability of the atom to be in its excited state and the probability of it to be in its ground state. It can be calculated as the expected value of the operator $\hat{\sigma}_z$, namely, $W(t) = \langle \psi(t) | \hat{\sigma}_z | \psi(t) \rangle$. From (19), we get

$$W(t) = \sum_{n=0}^{\infty} P_n(\gamma) \left\{ \cos^2(\Omega_{n+1} t) + \left[ \frac{\Delta^2}{4} - g^2(n+1) \right] \frac{\sin^2(\Omega_{n+1} t)}{\Omega_{n+1}^2} \right\}, \tag{20}$$

where $P_n$ is the probability of detecting $n$ photons in the field, which is given by the Poisson distribution

$$P_n(\gamma) = e^{-|\gamma|^2} \frac{|\gamma|^{2n}}{n!}, \tag{21}$$

with $\gamma = \beta + \alpha$, and $\alpha$ is given in (11). Besides,

$$\Omega_n = \left( \frac{\Delta^2}{4} + g^2 n \right)^{1/2}. \tag{22}$$

It is important to note that the expression (20) differs from that of the conventional JCM only in a shift in the probability distribution of photons: we go from $P_n(\beta)$ in the usual JCM, to

$P_n(\beta + \alpha)$ in the driven system. Then, the magnitude of the shift is determined by the couplings $\zeta$ and $g$. Fig. 2 illustrates the aforementioned effect; Fig. 2(a) and Fig. 2(b) show the atomic inversion $W(t)$ in the driven and conventional JCM, respectively. For the chosen values of the parameters, it is evident that the occurrence time of the first revival in the driven case [Fig. 2(a)] increases with the value of $\alpha$, in comparison with the conventional JCM [Fig. 2(b)]. In other words, as $\zeta$ ($g$) is increased (decreased), there is an observed displacement in time at which the first revival occurs; from a physical point of view, this means that if the coupling $\zeta$ between the classical field and the atom is far greater than that between the atom and the quantized field $g$, the transitions may be suppressed by the interaction with the classical field.

## 4.2 Average photon number

It is crucial to analyze another observable: the expectation value of the number operator $\hat{n}$, namely $\langle \hat{n}(t) \rangle = \langle \psi(t)| \hat{n} |\psi(t) \rangle$. By studying $\langle \hat{n}(t) \rangle$, we can get a better understanding of the statistical properties of the system, including the photon distribution and its relation with the dynamics of the atom-field interaction. From (19), we obtain

$$\langle \hat{n}(t) \rangle = S_1(t) - 2\alpha \mathrm{Re}\left[\gamma \exp(-\mathrm{i}\Delta_c t) S_2(t)\right] + \alpha^2, \tag{23}$$

where

$$S_1(t) = \sum_{n=0}^{\infty} P_n(\gamma)\left[|\gamma|^2 \left|V_1^{(n+2)}(t)\right|^2 + (n+1)^2 g^2 \left|V_2^{(n+1)}(t)\right|^2\right], \tag{24}$$

$$S_2(t) = \sum_{n=0}^{\infty} P_n(\gamma)\left[\bar{V}_1^{(n+1)}(t) V_1^{(n+2)}(t) + (n+2) g^2 V_2^{(n+1)}(t) V_2^{(n+2)}(t)\right], \tag{25}$$

and

$$V_1^{(n)}(t) = \cos(\Omega_n t) - \mathrm{i}\frac{\Delta}{2}\frac{\sin(\Omega_n t)}{\Omega_n}, \tag{26}$$

$$V_2^{(n)}(t) = \frac{\sin(\Omega_n t)}{\Omega_n}, \tag{27}$$

with the bar denoting complex conjugation, and $\mathrm{Re}[z]$ meaning the real part of $z$. It is relevant to note that similar to the case of the atomic inversion (20), the probability distribution of the number of photons in (23) undergoes a change when going from the conventional to the driven JCM: $P_n(\beta) \to P_n(\beta + \alpha)$. Nevertheless, the resulting $\langle \hat{n}(t) \rangle$ reveals a modification of the Rabi frequency $\Omega_n$ (this can be particularly seen in the shifts of the labels $\Omega_n \to \Omega_{n+1}$, $\Omega_n \to \Omega_{n+2}$ in the expressions for $S_1(t)$ and $S_2(t)$ above, through $V_1^{(n)}(t)$ and $V_2^{(n)}(t)$), which leads to a strikingly different behavior of the average photon number, in comparison to the standard JCM. This fact is illustrated in Fig. 3; in Fig. 3(a) the average photon number $\langle \hat{n}(t) \rangle$ in the driven JCM is shown, and in Fig. 3(b) the average photon number of the usual JCM is depicted; the same values of the parameters used in Fig. 2 are employed. Unlike the usual JCM [Fig. 3(b)], in which $\langle \hat{n}(t) \rangle$ exhibits a behavior similar to the atomic inversion [Fig. 2(b)], the driven JCM shows a completely different dynamics due to the direct influence of the external classical field on the cavity mode that feeds it with photons.

In addition, in the driven JCM the average photon number shows the *super revivals* discussed in [27]; if we analyze $\langle \hat{n}(t) \rangle$ at times larger than those of Fig. 3, it can be observed [Fig. 4(a)] that $\langle \hat{n}(t) \rangle$ shows a collapse, just as in the case of the conventional JCM [Fig. 3(b)], though at larger times. Moreover, at even larger times [see Fig. 4(b)], the corresponding revival can be appreciated. Such large-scale fluctuations (referred to as super revivals) were previously noted and studied in [27], where the external classical field drives the atom only.

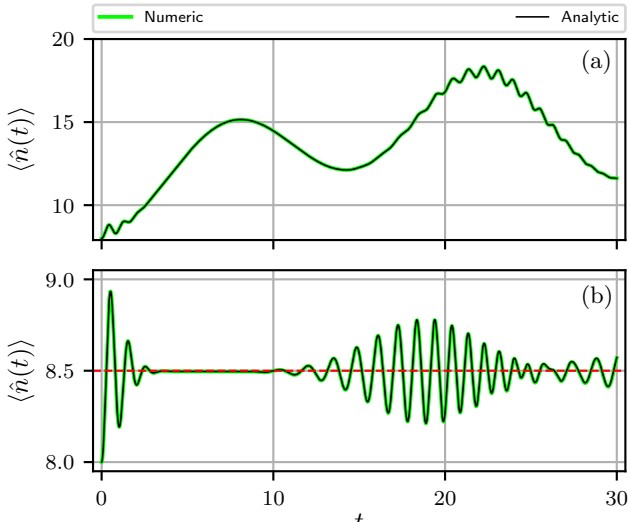

Figure 3: Average photon number $\langle \hat{n}(t) \rangle$ corresponding to the same initial condition and parameters used in Fig. 2. In (a) and (b), the average photon number is shown for the driven JCM and the standard JCM, respectively. The black lines correspond to the analytical result, while the green ones represent the numerical result.

Thus, they are present in the general case as well, where the classical field drives also the quantized cavity field, as can be clearly seen from Fig. 4.

Finally, it can be appreciated that the collapse in Fig. 4(a) occurs at $\langle \hat{n} \rangle \sim 13.44$ (red dashed line), while in the conventional case [Fig. 3(b)] it does occur at $\langle \hat{n} \rangle = 8.5$ [35]. Of course, this is attributed to the classical driving field that, as mentioned, provides the quantized one with photons, increasing $\langle \hat{n}(t) \rangle$.

### 4.3 Mandel $Q$ parameter

The Mandel $Q$ parameter is defined as follows [39]:

$$Q = \frac{\langle \hat{n}^2 \rangle - \langle \hat{n} \rangle^2}{\langle \hat{n} \rangle} - 1 \,, \tag{28}$$

and it measures the deviation from a Poissonian distribution. In other words, it gives information about the nature (sub- or super-Poissonian) of the quantized cavity field. For $Q > 0$ ($Q < 0$) we have a super (sub) Poissonian distribution of photons; for $Q = 0$, we have a Poissonian distribution of photons. In Figure 5, we show the Mandel $Q$ parameter as defined by (28) for the driven JCM, as well as for the conventional case. It is seen that in both, the driven and the conventional JCM, the field shows sub- and super-Poissonian features. In particular, the driven case (a) presents a slower dynamics, which is in agreement with what has been observed in the case of the average photon number $\langle \hat{n}(t) \rangle$, due to the presence of the driving classical field.

### 4.4 Entanglement and von Neumann entropy

Even when the quantized field and the atom are initially separate entities, while interacting they become together a composed system; in other words, the initial separable state becomes mixed. An accurate quantitative measure of the degree of mixing is obtained from the (von Neumann) entropy of the system, defined as [35]

$$S = -\mathrm{Tr} \{ \hat{\rho} \ln \hat{\rho} \} \,, \tag{29}$$

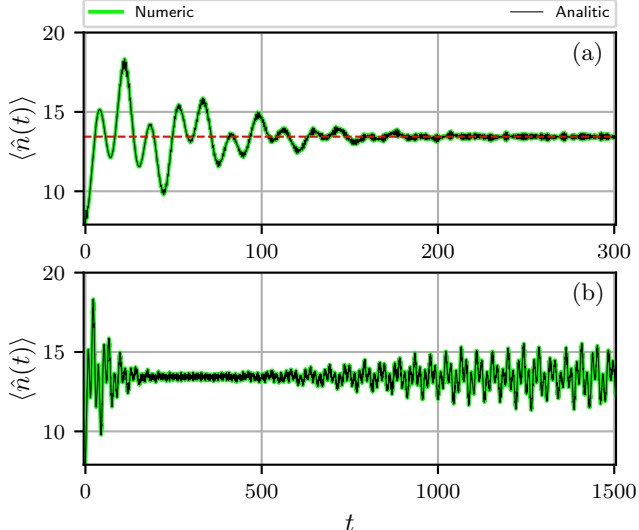

Figure 4: Average photon number $\langle \hat{n}(t) \rangle$ in the driven JCM for relatively large time. The same initial condition and parameters of Fig. 2 were used. In (a) the collapse of $\langle \hat{n}(t) \rangle$ can be clearly appreciated, while in (b) the collapse-revival is observed. The black lines denote the analytical result, while the green ones represent the numerical result.

with $\hat{\rho} = |\psi(t)\rangle \langle \psi(t)|$ the density matrix of the composed system.

As the initial states of the field and atom are pure states ($|\psi(0)\rangle = |\beta, e\rangle$), the corresponding initial entropies of the quantum field and the atomic subsytems, $S_F$ and $S_A$, are equal to zero. In fact, as the initial state of the composed system is a pure (separable) state, the total entropy $S$ is zero as well. Furthermore, as the entropy of a closed system does not change in time, we have $S(t) = 0$ for all $t$. From the Araki-Lieb theorem [2, 35, 36]

$$|S_A - S_B| \leq S \leq S_A + S_B \,, \tag{30}$$

it follows that $S_F(t) = S_A(t)$. Therefore we can focus, for instance, on the entropy of the atomic subsystem

$$S_A = -\text{Tr}_A \{\hat{\rho}_A \ln \hat{\rho}_A\} \,, \tag{31}$$

where

$$\hat{\rho}_A = \text{Tr}_F \{\hat{\rho}\} \,. \tag{32}$$

Using basic properties of the trace, it is easy to show that the atomic entropy (31) is given by

$$S_A = -\lambda_1 \ln \lambda_1 - \lambda_2 \ln \lambda_2 \,, \tag{33}$$

with $\lambda_1$ and $\lambda_2$ the eigenvalues of the matrix $\hat{\rho}_A$.

Fig. 6 shows a comparison of the atomic entropy for the driven and standard JCM. It can be observed that, in the driven case [Fig. 6(a)] the minimum entropy is displaced to larger values of $t$ (dashed red vertical line, at around $t \sim 11.5$), with respect to the minimum entropy (at around $t \sim 9.83$) in the conventional JCM [Fig. 6(b)]. This in turn corresponds to the displacement observed in the atomic inversion (Fig. 2) caused by the classical driving field. At the minimum entropy, the quantum field and two-level atom subsystems behave nearly as separate independent entities (see [36] and references therein). Also, the decreasing of the entropy to its minimum is known to coincide with the collapse of $W(t)$ (see for instance [35] and compare time scales in Fig. 2 and Fig. 6). In other words, in the driven case [Fig. 6(a)] the

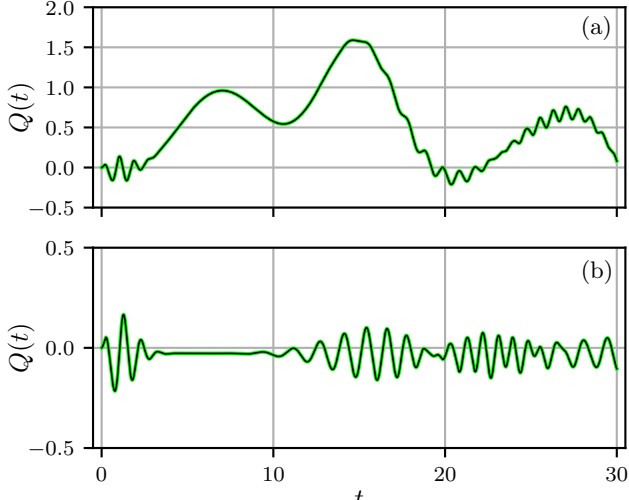

Figure 5: Mandel $Q$ parameter (28) in the driven JCM (a), and its comparison with the conventional case (b). The same initial condition and parameters of Fig. 2 were used. These plots correspond to numerical calculations, however the analytical expression is pretty straightforward to be obtained from the exact solution (6).

entropy takes longer time to reach its minimum value, in comparison with the conventional case [Fig. 6(b)]. This is as well in agreement with the longer collapse observed in Fig. 2(a), in comparison with Fig. 2(b), due to the classical driving field.

## 5 Dispersive model

In this section we analyze the dispersive interaction for the general driven case, this means that $|\Delta| \gg g$ is considered. In turn, the importance of the dispersive regime lays on a couple of facts. First, in such a regime the solution $|\psi(t)\rangle$ of the Schrödinger equation (6) simplifies considerably due to the condition $|\Delta| \gg g$. On the other hand, and more importantly, the dispersive regime is quite useful to obtain, both theoretically and experimentally, highly non-classical (entangled) states of light [7,36] (see also [24,25] and references therein). Particularly, Schrödinger cat states can be straightforwardly constructed, as we shall show in the following.

Starting from (9), with $\hat{\mathcal{H}}_D = \hat{\mathcal{H}}_{\text{JCM}}$, we propose the infinitesimal rotation $|\chi(t)\rangle = R^\dagger |\varphi\rangle$, with $\hat{R} = \exp[\mu(\hat{\sigma}_+\hat{a} - \hat{\sigma}_-\hat{a}^\dagger)]$, and $\mu \ll 1$ a parameter to be determined. The effective (diagonal) Hamiltonian $\hat{\mathcal{H}}_{\text{eff}} = \hat{R}\hat{\mathcal{H}}_{\text{JCM}}\hat{R}^\dagger$ is then

$$\hat{\mathcal{H}}_{\text{eff}} = \left(\Delta_c + \frac{2g^2}{\Delta}\hat{\sigma}_z\right)\hat{n} + \left(\Delta_{eg} + \frac{2g^2}{\Delta}\right)\frac{\hat{\sigma}_z}{2} + \frac{g^2}{\Delta}, \tag{34}$$

where we have set $\mu = \frac{g}{\Delta}$, and dropped all the powers of $\mu$ greater than two; this is deeply related to what is known as *small rotations* approach [40]. Consequently, the state vector is

$$|\psi(t)\rangle = \hat{T}^\dagger \hat{D}^\dagger(\zeta/g)\hat{R}^\dagger \hat{U}_{\text{eff}}(t)\hat{R}\hat{D}(\zeta/g)|\psi(0)\rangle, \tag{35}$$

where $\hat{U}_{\text{eff}} = e^{-\mathrm{i}t\hat{\mathcal{H}}_{\text{eff}}}$.

In this section a more general initial condition is considered for the atom: a superposition of the excited and ground states [7]

$$|\psi(0)\rangle = |\beta\rangle \otimes \frac{1}{\sqrt{2}}\left(|e\rangle + e^{\mathrm{i}\phi}|g\rangle\right), \tag{36}$$

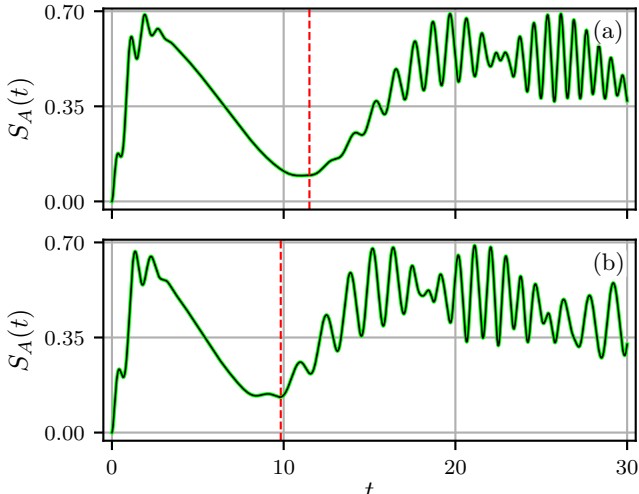

Figure 6: Entropy $S_A(t)$ corresponding to the same initial condition and parameters used in Fig. 2. In (a) and (b) the entropy is shown for the driven and standard JCM, respectively. These plots correspond to numerical calculations, however the analytical expression is pretty straightforward to be obtained from the exact solution (6), (see also Ref. [35]).

with $0 \le \phi < 2\pi$. Besides, as we have supposed that $\mu \ll 1$, we can take $\hat{R} \approx \hat{1}$. After some straightforward calculations, the state $|\psi(t)\rangle$ of the system is found to be

$$|\psi(t)\rangle = \frac{e^{-i\Theta}}{\sqrt{2}} \Big[ \Lambda e^{i\alpha \operatorname{Im}(\kappa_+)} |(\kappa_+ - \alpha)e^{-i\omega_0 t}\rangle \otimes |e\rangle + e^{i\phi}\bar{\Lambda}e^{i\alpha \operatorname{Im}(\kappa_-)} |(\kappa_- - \alpha)e^{-i\omega_0 t}\rangle \otimes |g\rangle \Big],$$
(37)

where $\Theta = \frac{g^2 t}{\Delta} + \alpha \operatorname{Im}(\beta)$, $\Lambda = \exp[-i\,t(\frac{\omega_{eg}}{2} + \frac{g^2}{\Delta})]$, $\kappa_\pm = (\beta + \alpha)\exp[-i\,t(\Delta_c \pm \frac{2g^2}{\Delta})]$, and $\operatorname{Im}(z)$ denotes the imaginary part of $z$. The state in (37) is an entangled (highly mixed) state of the quantum field and the two-level atom. It is a generalization of the Schrödinger cat states studied in sections 4.8 and 10.5 of Ref. [7]. Also, the initial condition (36) can be easily implemented experimentally, as explained in [7]. Moreover, despite the more involved initial condition used in this section, it is pretty straightforward to obtain the rather simple expression (37) in the large detunning limit $\Delta \gg g$.

# 6 Conclusions

Using an invariant approach as a preamble, we have shown that the theoretical driven Jaynes-Cummings model can be transformed into the standard one through a pair of unitary transformations. This in turn allows to obtain the exact analytical solution of the Schrödinger equation for the driven system, as well as the corresponding dynamical variables. Some examples of interest were given: atomic inversion, average of the number of photons in the electromagnetic field, and subsystem entropies. The classical driving field has shown to have notorious effects in such dynamical variables.

The present work constitutes then a generalization of the method established in [27], where it is considered that the classical field drives the atomic system only. Also, it represents a further step with respect to Ref. [26], as our solution serves to obtain the dynamical variables of the driven system in a straightforward manner, a result that is not present in the approach in [26]. In addition, the atom in the studied driven system is susceptible to be used as a catalyst, as described in Ref. [41], this will be reported elsewhere.

In all the cases, the analytical results had proven to be in good agreement with the numerical calculations. Also, they reduce to those of the standard Jaynes-Cummings model in the appropriate limit.

# Acknowledgments

**Funding information** I. Bocanegra acknowledges CONAHCyT (México) for financial support through the postdoctoral fellowship 711878 and projects A1-S-24569 and CF19-304307, also to IPN (México) for supplementary economical support through the project SIP20232237 and the Spanish Ministry of Science and Innovation (MCIN) with funding from the European Union Next Generation EU (PRTRC17.I1) and the Department of Education of Castilla y León (JCyL) through the QCAYLE project, as well as MCIN projects PID2020-113406GB-I00 and RED2022-134301-T. In turn, L. Hernández Sánchez thanks the Instituto Nacional de Astrofísica, Óptica y Electrónica (INAOE) and the Consejo Nacional de Humanidades, Ciencias y Tecnologías (CONAHCyT) by the PhD scholarship awarded (No. CVU: 736710).

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
