# Peer review of "Invariant approach to the Driven Jaynes-Cummings model"

_SciPost Physics, doi:SciPost Phys. 16, 007 (2024)_

## Round 3 · Referee Report · José F. Récamier (Referee 1) · 2023-11-15

Report

The authors have taken care of all my questions and I consider that the manuscript should be published as it is now.

---

## Round 3 · Referee Report · Anonymous (Referee 2) · 2023-12-12

Report

The revised manuscript has successfully addressed all changes and amendments suggested in the first report.

Especially the addition of section IV.C on the Mandel Q parameter is a very valuable addition.

Publication of the manuscript is now recommended without further reservations.

---

## Round 3 · Author Response

Reviewer: 1 1. Reviewer’s Comment:I consider that the manuscript is well written and contains interesting results so it could be published as it is. There are some minor points to be taken into consideration: The figures present analytic and numerical results, since the analytic results are exact I do not see the reason for including purely numerical results. 2. Reviewer’s Comment:Weakness: The last section in the manuscript does not seem to provide much information, I think its relevance could be stated more clearly. Requested changes: Improve the last section of the manuscript.

Reviewer: 2 1. Reviewer’s Comment:The major change I suggest is in the title of the submission which I find to be too general. Since the manuscript deals exclusively with the driven Jaynes-Cummings model, not the atom-field interaction in general, I suggest that this is made clear already in the title by, e.g., changing “atom-field interaction” into “Jaynes-Cummings model”. 2. Reviewer’s Comment:Due to developments in recent decades, a model going beyond the rotating wave approximation, the quantum Rabi model, also introduced by Jaynes and Cummings in the article cited of the present manuscript, has attracted considerable experimental and theoretical attention. The applicability of the quantum Rabi model is not restricted, as in the Jaynes-Cummings, to small atom-field interaction strength. It would therefore be appropriate to mention this model in the introduction of the present paper together with some key references. For example, in Niemczyk T et al 2010 Circuit quantum electrodynamics in the ultrastrong-coupling regime Nat. Phys. 6 772 and Casanova J, Romero G, Lizuain I, García-Ripoll J J and Solano E 2010 Deep strong coupling regime of the Jaynes-Cummings model Phys. Rev. Lett. 105 263603 larger values of the atom-field interaction strength have been explored. While the Jaynes-Cummings model can be solved by elementary means, this is not the case for the quantum Rabi model which has only been solved quite recently in Braak D 2011 Integrability of the Rabi model Phys. Rev. Lett. 107 100401 using sophisticated analytical techniques. 3. Reviewer’s Comment:A minor typo: thing → think in the fourth line of section II. 4. Reviewer’s Comment:The explanation that Ωn is now a number, not anymore an operator ˆΩn as in equations (17) and (18), should already appear earlier, after equation (20). Answer: Thanks a lot, we completely agree that such precision is necessary. As you can now appreciate, the expression (22) has been added, from which we hope that the previous ambiguity is completely avoided. 5. Reviewer’s Comment:An interesting quantity not calculated by the present authors, but presumably within the reach of the method presented, would be the variance of the average photon number. 6. Reviewer’s Comment:For clarity, the inset indicating the analytically calculated versus the numerically obtained results in figure 5 should also appear. 7. Reviewer’s Comment:After equation (26), it is said that the Rabi frequency Ωn is modified, but no explicit expression is given. Do the authors refer back to equation (18)? Is this a statement with reference to figure 3? 8. Reviewer’s Comment:There are a few awkward phrases which should be taken care of, e.g. −as the operator has been already acted, −being ˆn = ˆa†ˆa, the latter appearing similarly in several places.

---

## Round 3 · List of Changes

Reviewer: 1 1. Answer: Thanks a lot for the interest. We would like to comment that the numerical calculations where performed in order to validate the analytical expressions. Once we are convinced that our general solution is correct, and gives correct analytical expressions, more and more dynamical variables of the system can be straightforwardly analyzed using, for instance, numerical calculations, as the analytical expressions are sometimes awfully large and the calculations cumbersome. In this way, we can focus more on the physics behind the dynamics of the system and less in the mathematical aspects of it. Such is the case, for instance of the section IV, subsection D where we analyze the entropy of the system, as well as the (recently added, for request of one of the reviewers) subsection C, in the same section. 2. Answer: We are very grateful for the suggestion you make addressed to highlight the importance of the last section. By keeping that in mind, we have added some lines at the beginning of the section in order to establish clearly the relevance of the dispersive regime from both theoretical and experimental points of view. The corresponding references are also cited. Furthermore, a couple of supplementary references have been added (References [24] and [25]), in order to emphasize the applicability of the model in such a regime; such references accentuate principally the experimental realization. From this, we hope, the reader will be fully convinced of the relevance of the dispersive interaction in both the driven Jaynes-Cummings system, as well as in the conventional model.

Reviewer: 2 1.Answer: We all thank a lot for the suggestion and we consider that, indeed, referring to the “Jaynes-Cummings model” is more accurate than the more general “atom-field interaction”. We have made the corresponding modification, as you can appreciate, the new title of the manuscript is "Invariant approach to the driven Jaynes-Cummings model". 2. Answer: We thank a lot for the suggestion and also for bring to our attention such important references about the (more general) Rabi model. As you can observe, we have added a some lines in the introduction making reference to the Rabi model. Besides, the References you have gently shared with us have been added as well (Ref. [3-5]). 3. Answer: Thanks a lot for bring our attention to such a mistake. As you can appreciate now, the corresponding change has been made: in the sentence now appears "think" instead of "thing". 4. Answer: Thanks a lot, we completely agree that such precision is necessary. As you can now appreciate, the expression (22) has been added, from which we hope that the previous ambiguity is completely avoided. 5. Answer: Thanks a lot for the suggestion. Indeed, it is possible to obtain the variance of the average photon number by means of the method developed in the manuscript. Furthermore, the Mandel Q parameter giving information about the nature of the statistics of the electromagnetic field in the cavity, and where the variance is implicit, can be obtained as well. As you can appreciate, the subsection IV. C. has been added in order to analyze such an important physical quantity. We are in debt with you for such a valuable suggestion. 6. Answer: Thanks again. We would like to clarify that the calculation giving rise to Figure 5 has been performed numerically. That is the reason such a statement did not appear into the corresponding caption. However, as you can appreciate now, we have added a comment stating clearly that the plot is obtained by means of numerical calculations, and also cited the corresponding reference in case the reader is interested in obtaining the result in a purely analytical way. 7. Answer: Thanks again for the interest, and for helping us to clarify this point. Actually, we refer to the fact that in ⟨n(t)⟩, given by (23), the Rabi frequency presents shifts Ω_n → Ω_{n+1}, Ω_n → Ω_{n+2}, which can be appreciated in both S_1(t) and S_2(t) (expressions (24) and (25)), through V _1^{(n+2)} (t), V _2^{(n+1)} (t), etc. ((26) and (27)). Also it is worth to remark that, precisely as you mentioned, the effect of such shifts is shown in Figure 3. So, in that sense it is indeed a statement with reference to that Figure. However, we have made the corresponding comment, as you can already appreciate, in order to be as clear as possible with such a statement. 8. Answer: Thanks a lot for pointing out that such phrases are unintelligible. The first one has been removed, as it was making reference precisely to the point indicated in the Reviewer’s Comment no. 4. In turn, regarding the second phrase cited, it can be observed that the word "being" has been replaced with some other (more precise) words as "where", "with", etc., according to the context.

---

## Editorial Decision

published